# Parenting and childhood obesity: Validation of a new questionnaire and evaluation of treatment effects during the preschool years

**Maria Somaraki[1], Anna Ek[2], Karin Eli[3], Sofia Ljung[4], Veronica Mildton[1], Pernilla Sandvik[1], Paulina Nowicka**[1,2] *

1 Department of Food Studies, Nutrition and Dietetics, Uppsala University, Uppsala, Sweden, 2 Division of Pediatrics, Department of Clinical Science, Intervention and Technology, Karolinska Institutet, Stockholm, Sweden, 3 Division of Health Sciences, Warwick Medical School, University of Warwick, Coventry, United Kingdom, 4 Aleris Rehab, Skärholmen, Sweden

* Paulina.nowicka@ikv.uu.se

## Abstract

### Objectives

Parenting is an integral component of obesity treatment in early childhood. However, the link between specific parenting practices and treatment effectiveness remains unclear. This paper introduces and validates a new parenting questionnaire and evaluates mothers' and fathers' parenting practices in relation to child weight status during a 12-month childhood obesity treatment trial.

### Methods

First, a merged school/clinical sample (n = 558, 82% mothers) was used for the factorial and construct validation of the new parenting questionnaire. Second, changes in parenting were evaluated using clinical data from the More and Less Study, a randomized controlled trial (RCT) with 174 children (mean age = 5 years, mean Body Mass Index Standard Deviation Score (BMI SDS) = 3.0) comparing a parent support program (with and without booster sessions) and standard treatment. Data were collected at four time points over 12 months. We used linear mixed models and mediation models to investigate associations between changes in parenting practices and treatment effects.

### Findings

The validation of the questionnaire (9 items; responses on a 5-point Likert scale) revealed two dimensions of parenting (Cronbach's alpha $\geq$0.7): setting limits to the child and regulating one's own emotions when interacting with the child, both of which correlated with feeding practices and parental self-efficacy. We administered the questionnaire to the RCT participants. Fathers in standard treatment increased their emotional regulation compared to fathers in the parenting program (p = 0.03). Mothers increased their limit-setting regardless of treatment allocation (p = 0.01). No treatment effect was found on child weight status through changes in parenting practices.

**Data Availability Statement:** All relevant data are within the paper and its Supporting Information files.

**Funding:** PN was supported to conduct the study by the Swedish Research Council (2014-02404), Karolinska Institutet Doctoral Funds, the Swedish Society of Medicine, VINNOVA (2011-03443), Jerring Foundation, Samariten Foundation, Magnus Bergvall Foundation, Ingrid and Fredrik Thuring Foundation, Helge Ax:son Foundation, Crown Princess Lovisa Foundation, Foundation Frimurare Barnhuset in Stockholm, Foundation Pediatric Care, Foundation Martin Rind, Jane and Dan Olsson Foundation, Sigurd and Elsa Golje Memory Foundation, and iShizu Matsumurais Donation. The funding sources had no role in the study design, collection, analysis or interpretation of the data, writing the manuscript, or the decision to submit the paper for publication.

**Competing interests:** The authors have declared that no competing interests exist.

## Conclusion

Taken together, the findings demonstrate that the new questionnaire assessing parenting practices proved valid in a 12-month childhood obesity trial. During treatment, paternal and maternal parenting practices followed different trajectories, though they did not mediate treatment effects on child weight status. Future research should address the pathways whereby maternal and paternal parenting practices affect treatment outcomes, such as child eating behaviors and weight status.

## Introduction

Parents are integral to obesity treatment during the preschool years [1, 2]. Treatment approaches focus on promoting children's health behaviors, with the aim of improving children's energy balance and weight status [3, 4]. Thus, healthy eating and exercise are central components of treatment. Nevertheless, these behaviors need to be practiced in a supportive family environment, which is strongly influenced by parenting practices [1, 5].

Parenting is commonly conceptualized according to two dimensions, demandingness and responsiveness [6], which identify four parenting styles (authoritative, authoritarian, neglectful, and indulgent/permissive). These styles account for unique combinations of high and low endorsement of each dimension in relation to the other one [7]. Demandingness identifies parents' control and provision of structure, while responsiveness identifies parents' consideration of a child's needs. The parenting style described as authoritative, which ranks high in both dimensions (demandingness and responsiveness), has consistently been associated with favorable child behaviors and health outcomes including a healthy weight gain [8, 9]. However, most studies have focused on mothers, while the role of fathers remains largely unexplored [9]. This is a concern in light of fathers' unique contributions to their children's health [10, 11]. In particular, longitudinal data from Australia suggest that higher paternal responsiveness is associated with increased risk for obesity in early childhood, while similar associations have not been identified for any dimension of maternal parenting [12]. Therefore, a comprehensive assessment of parenting in relation to childhood obesity should take into account both parents.

While active parental involvement is required for effective obesity treatment in childhood [1, 2], little is known about how parents facilitate child weight loss [5, 13, 14]. Effective treatment programs integrate essential education on nutrition and physical activity with guidance for parents on how to maintain children's healthy behaviors [15–17]. This parent-focused approach is especially relevant during the preschool years, when parents have the greatest capacity to implement changes in the home environment and affect child weight status [18]. Indeed, obesity treatment among younger children is associated with clinically significant weight loss [19–21]. Reinehr et al. [19] in Germany evaluated an intensive 1-year long treatment program (without a control condition), which included parenting sessions and aimed to modify lifestyle factors. The study demonstrated that younger children (4–7 years old) maintained their weight loss after 4 years of follow-up to a greater degree than older children and adolescents. To explain these findings, the authors pointed to the fact that parents of younger children were the recipients of the program and bore the main responsibility for the implementation of lifestyle changes. Yet, well-designed randomized controlled trials (RCTs) for childhood obesity in the preschool years are still lacking, and research on programs that address parenting is particularly scarce [21, 22].

The present paper attempts to address this gap in the literature by evaluating parenting practices and their effects on child weight status during a 12-month follow-up of an RCT for obesity treatment among preschoolers, the More and Less study (ML study) in Sweden [23]. The ML study compares a parenting program with and without booster sessions (follow-up phone calls provided after the program, averaging 4 phone calls per family) and standard treatment; the primary study outcome is change in child weight status (body mass index standard deviation score, BMI SDS). The primary findings at 1-year post-baseline showed a greater decrease in child weight status among families randomized to the parenting program with booster sessions, compared to standard care and the parenting program without booster sessions [24]. In addition, the ML study includes a diverse sample with a high proportion of parents reporting foreign background (parent and/or their parents born outside Sweden) and lower educational attainment, unlike the majority of studies in the field which have included homogeneous samples [25]. Parental foreign background moderated treatment effects; specifically, boosters were necessary for sustained treatment effects among children whose parent(s) had foreign background [24]. The parenting program draws on the Oregon Model of Behavior Family Therapy, which emphasizes a core set of evidence-based parenting skills, i.e. encouragement, monitoring, limit-setting strategies, positive involvement, problem solving, and emotional regulation [26–28]. Thus, the parenting program focuses on important aspects of demandingness (e.g. limit setting) and responsiveness (e.g. emotional regulation), which compose the favorable authoritative parenting style with regard to childhood obesity [9, 29]. While parenting programs unrelated to obesity have addressed child behaviors in randomized studies successfully [30, 31], they have rarely been applied in the field of early childhood obesity [21]. The relevance of parenting skills in early childhood obesity, however, has been highlighted in observational studies [32, 33]. In addition to the parenting practices, the ML program offers developmentally appropriate information around healthy lifestyles (nutrition and physical activity) [23]. To evaluate the central parenting components of the ML program (encouragement, monitoring, limit-setting strategies, positive involvement, problem solving, and emotional regulation) a valid user-friendly tool was required. Given the lack of appropriate evaluation instruments, we developed a questionnaire to assess changes in parental behaviors after participating in the ML program.

## Aims

This paper has a two-fold aim. First, to validate a new questionnaire assessing parenting practices (study I) and second, to evaluate the change in parenting practices, as secondary outcomes, in relation to childhood obesity treatment for preschoolers (study II).

The specific objectives are:

1. to validate a questionnaire on key parenting practices addressed during the ML program *(study I)*;

2. to evaluate changes in parenting practices during treatment *(study II)*;

3. to examine whether changes in parenting practices mediate changes in child weight status during treatment *(study II)*.

## Hypotheses

**Study I—Validation of a questionnaire on parenting practices.**   The questionnaire was developed to include specific items describing the parenting practices addressed in the ML parenting program. Thus, we expect that the items will cluster into factors reflecting

monitoring, positive involvement, limit setting, problem solving, emotional regulation, and encouragement. Moreover, we hypothesize that parenting practices assessed through the questionnaire will correlate with parent reported feeding practices (restricting, pressuring, and monitoring of child food intake) and problematic child behaviors in relation to food, physical activity and obesity. In addition, we expect that parenting practices will discriminate between children with and without obesity.

**Study II—Evaluation of parenting practices in obesity treatment.** Our hypotheses were informed by the primary findings of the ML study, which showed that children of families who were randomized to the parenting program–in particular, those who received the additional booster sessions–decreased their weight status more compared to children of families in standard treatment [23, 24]. Therefore, it is assumed that parents who participated in the parenting program with boosters will demonstrate a greater increase in effective parenting practices compared to parents in the other two conditions (parenting program without boosters and standard treatment) over the 12-month follow-up. In addition, it is hypothesized that changes in parenting practices will, at least partly, mediate the effect of treatment on changes in child weight status. Thus, among families randomized to the parenting program with boosters, changes in parenting practices should be on the causal pathway of treatment effects on child weight status.

## Materials and methods

This study focuses on the measurement and evaluation of the evidence-based parenting practices addressed in the ML program. During the program's weekly group sessions, parenting practices were introduced (i.e. encouragement, monitoring, limit setting strategies, positive involvement, problem solving, and emotional regulation) to help parents respond to child behaviors effectively, and to increase parental capacity to implement changes in the home environment that support children's healthy lifestyles [24]. However, to investigate whether parenting practices affected the clinically significant weight loss among children in the parenting group with boosters, a questionnaire was required to assess these parenting practices and capture changes therein [24].

### Study I

According to international standards for developing questionnaires, we applied a mixed methods approach to 1) develop and 2) validate the questionnaire in a systematic way [34, 35]. The development included the following stages: literature search and face validity (performed by the research group), content validity (consulting experienced professionals in child health care) and cognitive interviews (consulting parents). Once a pool of relevant items was constructed, the validation of the questionnaire consisted of the identification of patterns (sub scales or factors) between the questionnaire items. A detailed description of the different stages of the development of the new questionnaire is provided below:

**Literature search.** To identify papers describing parenting questionnaires, Google, Google Scholar, PubMed, Sirus and Web of Science were searched. Relevant keywords were:"Parenting","Parent/Parental", and"Child rearing" in combination with"skills","questionnaire","form", and"techniques". When questionnaires were identified, their corresponding authors were contacted with a request for a full item list. This resulted in the collection of 14 questionnaires and 396 items (in English and Swedish), which were deemed relevant to evaluating the practices addressed in the ML program.

**Face validity.** In several group meetings, three members of the research team (health care professionals with background in dietetics, pediatrics and psychology and with experience of

working with families of children with obesity) identified items relevant to the assessment of parenting. Items from existing questionnaires were retained if the study team deemed them relevant for assessing encouragement, monitoring, limit setting, positive involvement, problem solving, and emotional regulation. Duplicate items were excluded, resulting in a sharp reduction in the number of items. New items were devised based on expert opinion of health care professionals with clinical experience of working with children and their families. The process yielded 38 items categorized in four subscales (positive involvement, encouragement, limit setting, monitoring).

**Content validity.** Seven experts in child health care, pediatrics, and pediatric psychology rated the 38 items based on how well they reflected the parenting skills of interest. The respective Content Validity Indexes on the sub scale level (S-CVI) were computed [36]. Subscales were deemed to measure what they intended to (by the child health care experts) if they included enough items relevant for assessing aspects of the corresponding parenting practice, meaning that on the sub scale level, only items which the experts considered relevant were retained, contributing to the calculation of the S-CVI. Only monitoring (0.92) and limit setting (0.96) reached the recommended level (S-CVI>0.90) [36].

**Cognitive interviews.** To evaluate the content and wording of the items on monitoring and limit setting, cognitive interviews were conducted by three members of the research team [31]. Six parents of preschoolers were recruited through principals of two preschools in Stockholm. Following verbal probing, an interview technique commonly used in cognitive interviewing, parents were presented with the printed questionnaire and were prompted to think aloud about their answer [37]. Follow-up questions allowed parents to express their opinions on question wording and answer options. This process resulted in the following 12 items:

1. My child can make me change my mind to something I first said no to.

2. If my child and I don't agree on something, I wait with all discussion until the child has calmed down.

3. I think it is difficult to say no to my child.

4. How I handle my child's behavior depends on how I feel.

5. If my child and I disagree on something, we end up doing what my child wants.

6. If my child doesn't do what I say I find it hard to control my emotions.

7. I can change my mind if my child throws a tantrum over something I have decided.

8. My child listens to what I say.

9. If my child and I want different things, we end up falling out with each other.

10. I think it's easy to get my child to think about something else if s/he starts nagging about something.

11. If my child doesn't listen to me, I get frustrated.

12. I think it is hard to set limits to my child.

The response options for all items ranged from 1 to 5: '1 = Not at all', '2 = To a small extent', '3 = Somewhat agree', '4 = Agree', '5 = Agree completely'.

**Validation of the questionnaire on parenting practices.** *Recruitment*. To validate the questionnaire, parents' reports from two samples were used (a school sample and the clinical sample from the ML study) [23].

*School sample.* To reach the parents of 4–5 year olds, thirty preschools were selected; to reach the parents of 6 year olds, fifteen schools were selected. The preschools and schools were selected from different areas across Stockholm County with low, medium, and high prevalence of obesity [38]. Among those, twenty preschools and five schools agreed to participate; 931 parents received the new questionnaire on parenting, the Child Feeding Questionnaire (CFQ), the Lifestyle Behaviour Checklist (LBC) and a background questionnaire. A total of 431 parents returned completed questionnaires in a closed envelope [39, 40]. All data were collected anonymously.

*Clinical sample (ML study).* Baseline data from the ML study were used, i.e. the questionnaire on parenting, the CFQ, the LBC and information about sociodemographic background. Since the ML study is the focus of study II, its experimental and longitudinal design is described in the next section.

*Covariates/background questionnaires.* Child BMI SDS was based on parent-reported data (child height and weight) for the school sample and on measured data for the clinical sample. Calculations of the BMI SDS and the classification of children according to their obesity status (children with obesity and without obesity)–in line with the focus of this paper on childhood obesity per se–were based on the criteria by Cole & Lobstein [41]. Moreover, 18 children were classified in the underweight category, according to age- and gender-specific criteria for thinness among children (child weight status equivalent to BMI<17) [42], and they were included in the non-obesity category. Parents' heights and weights were self-reported and used to calculate parental Body Mass Index (BMI = $kg/m^2$). Parents also reported on their education level (further categorized into university degree or no university degree) and their country of birth. Moreover, data on child age and gender were parent-reported in the school sample and were made available upon referral of children in the clinical sample.

*Exploratory factor analysis, reliability & validity.* The structure of the 12-item questionnaire was tested using Exploratory Factor Analysis (EFA) to identify the patterns between items and the ways they relate to each other, forming separate subscales (or factors). The identified factors were further examined for validity and reliability. Two instruments were used for construct validation, the CFQ and the LBC. These function as criteria measures, and we expected them to correlate with the factors in the new questionnaire, confirming its validity to assess parenting. Both instruments have been translated to Swedish and validated in Sweden [39, 43]. The CFQ assesses key feeding practices that parents employ in order to influence their child's food intake: restriction of access to certain energy-dense foods, pressure to eat, and monitoring of food intake [44]. While restriction and pressure to eat represent controlling feeding strategies that relate to coercion, monitoring is a form of control reflecting positive aspects of structure and guidance. Feeding practices are embedded in and reflect parenting practices [45, 46]. The LBC assesses parental reports on child problematic behaviors in relation to eating, physical activity and overweight, along with parental confidence in handling those [47]. Because the LBC addresses specific obesity-related behaviors, and because parents endorse certain practices in relation to perceived problematic behaviors of the child [26, 39, 48], correlations between the new questionnaire and the LBC were expected.

## Study II

The ML study is a parallel open label RCT, evaluating the effectiveness of a parenting program for obesity treatment among preschoolers [23]. Families of preschoolers (between 4 and 6 years old) with obesity [41, 49] were eligible to participate if 1) the child did not have any chronic or developmental condition that could affect weight and height; 2) the child did not receive any other treatment for obesity; and 3) parents/caregivers had sufficient knowledge of

the Swedish language to participate in the parenting program's group sessions and fill out questionnaires.

Families were randomized to the three treatment conditions, as described below:

1. Standard treatment (ST): ST represents the usual care offered in outpatient pediatric clinics, based on the action plan for childhood obesity in Stockholm County [50]. It emphasizes lifestyle modifications in eating and physical activity. During the first visit families met with a pediatrician. In follow up visits, families met mainly with a pediatric nurse but also with a dietician, psychologist, physiotherapist or occupational therapist.

2. Parenting program with booster sessions (PGB): Parents attended 10 weekly group sessions, each built around a parenting component along with a lifestyle component [23, 24]. After the end of the program and up to 12 months post-baseline, parents continued to receive support in implementing the content of the program through monthly phone calls.

3. Parenting program without booster sessions (PGNB): Parents attended the 10 sessions of the parenting program. However, after the end of the program, parents did not receive monthly phone calls.

**Sample size.** The sample size calculation for the ML study was based on data from a treatment study in Germany [51]. On the basis of power calculations, 75 children were needed in each treatment (parent-only and ST adjusted for dropout) to detect a difference of 0.3 BMI z score (0.5 SD) with 85% power at 12 months' follow-up. The calculations included an adjustment for a dropout rate of 21%, based on data from a similar study of obesity treatment focusing on parenting [52]. The sample size calculation has been described previously [24].

**Measurements.** *Child BMI SDS.* Child height and weight were measured at baseline and 3, 6, and 12 months post-baseline. Child height was measured by trained health care professionals to the nearest 0.1 cm using a fixed stadiometer. Children were weighed to the nearest 0.1 kg wearing light clothing. BMI was calculated based on weight and height. The primary outcome of the RCT, BMI SDS, was computed based on age- and sex-specific reference data [41].

*Parenting practices.* The questionnaire on parenting, consisting of 12 items, was administered at baseline, and at 3–6- and 12-months post-baseline in the ML study. The findings from the validation study (study I) informed the choice of items to be included when evaluating changes in parenting practices across treatment groups during the 12-month follow-up.

*Covariates/background characteristics.* Background questionnaires were filled out by parents, who reported on family structure (birth order of the child and if the child lived with both parents or not). In addition, parents (mothers and fathers separately) reported their age, weight, and height (to calculate BMI), level of education (with/without university degree) country of birth and their parents' country of birth, which further informed the categorization of parents (mothers/fathers) according to their foreign background (parent and/or their parents born outside Sweden). Information on child's age and gender were provided upon referral from healthcare.

Fig 1 provides an overview of the samples in study I and study II.

## Statistical analyses

**Study I.** Differences between subsamples were examined using independent samples t-test for continuous variables and chi-squared test for categorical variables. Descriptive characteristics are presented using means (standard deviations) and n (%), for continuous and categorical variables, respectively.

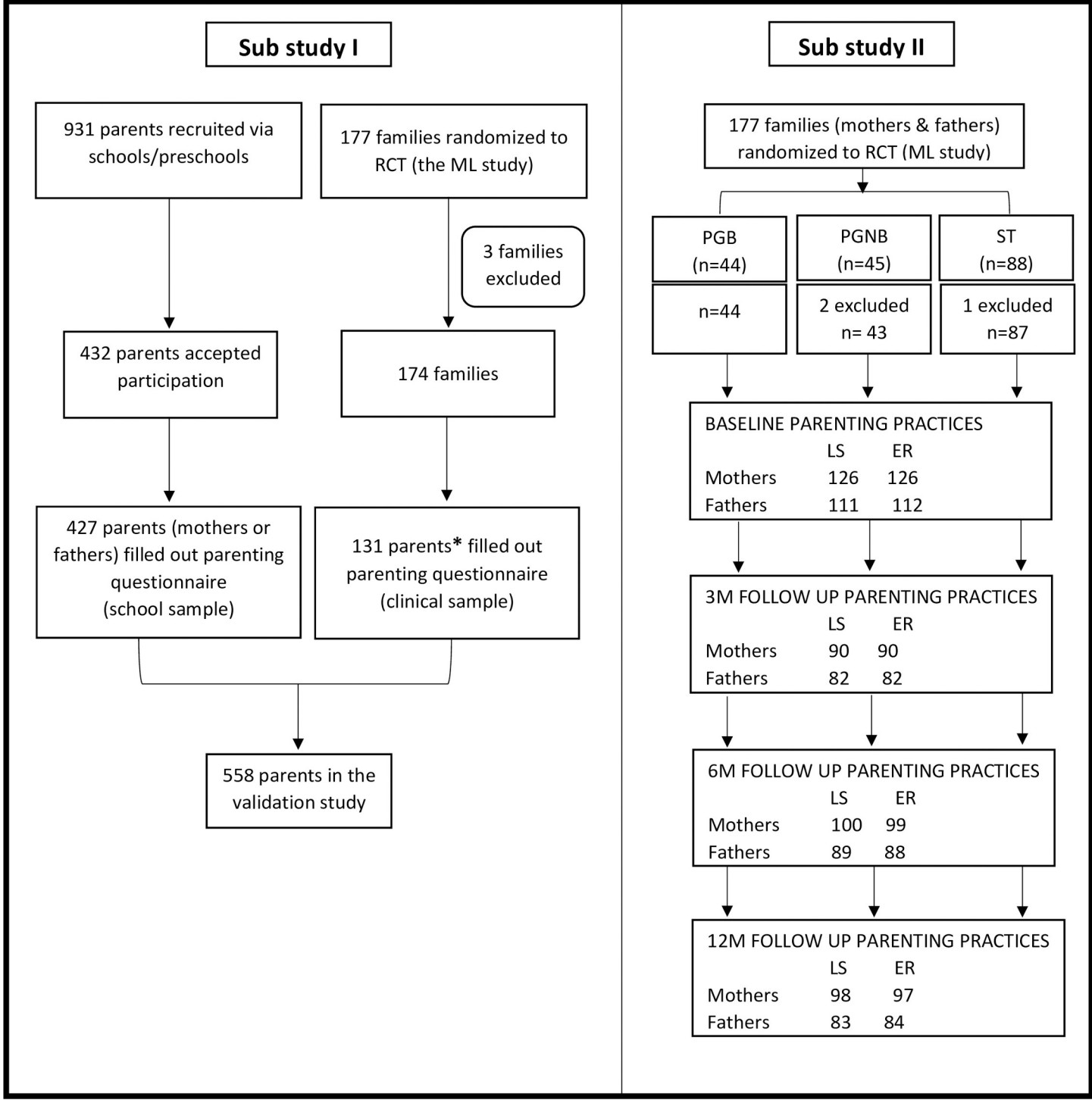

**Fig 1. The samples analyzed in the paper.** Left side: The two sub samples in the validation study (study I). Right side: sample in the ML study/ RCT with three treatment conditions (study II).* In study I, baseline data on parenting practices from the ML study were included. Only one parent's report on parenting was used in the validation study, this parent had also filled out a questionnaire about child eating behavior. The latter could have been filled out by one parent/caregiver only.

*EFA*. In total, 558 families (mothers or fathers) filled out the parenting questionnaire. EFA was applied to identify the parenting components assessed in the questionnaire. In particular, Principal Component Analysis (PCA) identified distinct factors underlying the 12 items along with the items that compose each factor. PCA with varimax normalized rotation (factors are not allowed to correlate) as well as with direct oblimin rotation (factors are allowed to correlate) was run on all 12 items, and the threshold for factor loading was set to 0.4. In addition, internal reliability coefficients (Cronbach's alpha) were calculated, informing optimal factor structure. Questionnaire items, except for three items, were worded to capture the opposite behavior of what evidence-based parenting practices would suggest (e.g., item 12 reads 'I think it is hard to set limits to my child'). Therefore, nine items were analyzed using reverse scoring. In all items, higher scores indicated higher levels of use of the practice described by the item. Based on the number of factors and the items composing each factor, the mean score of each factor was computed. This score was used for the subsequent analyses of validity. Higher mean scores related to higher endorsement levels of the respective parenting practice.

*Construct validation*. Nonparametric correlation coefficients (Spearman's rank correlation coefficient) were computed to quantify the strength and direction of associations between the parenting questionnaire and the CFQ and LBC. In addition, mean differences in parenting items between parents of children with obesity and parents of children without obesity were investigated using parametric t-tests (due to the ordinal nature of the parenting questions, the findings were confirmed using non-parametric Mann-Whitney U test and Wilcoxon signed rank sum test).

**Study II.** Background characteristics were compared across the treatment groups at baseline using one-way ANOVA (for continuous variables) and chi-squared test (for categorical variables).

Linear mixed models were used to evaluate the difference in treatment effects on evidence-based parenting practices. These models included the following variables: time (in months), treatment group (three conditions: PGB, PGNB and ST), and the treatment group-by-time interaction. The models also included random intercept and a random slope for time. The models were not adjusted for covariates due to the randomized design of the ML study. No additional procedures were applied to impute missing data at baseline or any follow-up time point. Estimated marginal means were computed based on the linear mixed models at baseline and 3-, 6- and 12-months post-baseline.

The PROCESS macro for SPSS, version 3.4.1 [53], was utilized to fit the mediation models and estimate the indirect effects of treatment on changes in child weight status through parenting practices (Fig 2), using 10 000 bootstrap samples to define Confidence Intervals (CIs) at the 95% level. In those models, changes in parenting practices (mothers' and fathers') were examined as potential mediators explaining treatment effects on changes in child weight status (the primary outcome in the ML study).

To obtain variables describing changes in mothers' and fathers' parenting practices and changes in child weight status the following process was followed [54]. Linear regression models were fitted for each parent and child in a long data format. The models included the parenting practices or child weight status outcome as the dependent variable and a continuous predictor for time (0, 3, 6, and 12 months) representing time since baseline. The computed slope (unstandardized b coefficient) for each individual reflects mean change per month in parenting practices and weight status for each child. These variables describe mean monthly change for each individual and they will be used in subsequent mediation models. No slope was calculated for individuals who had missing data at more than two time points. No imputation of missing data was conducted.

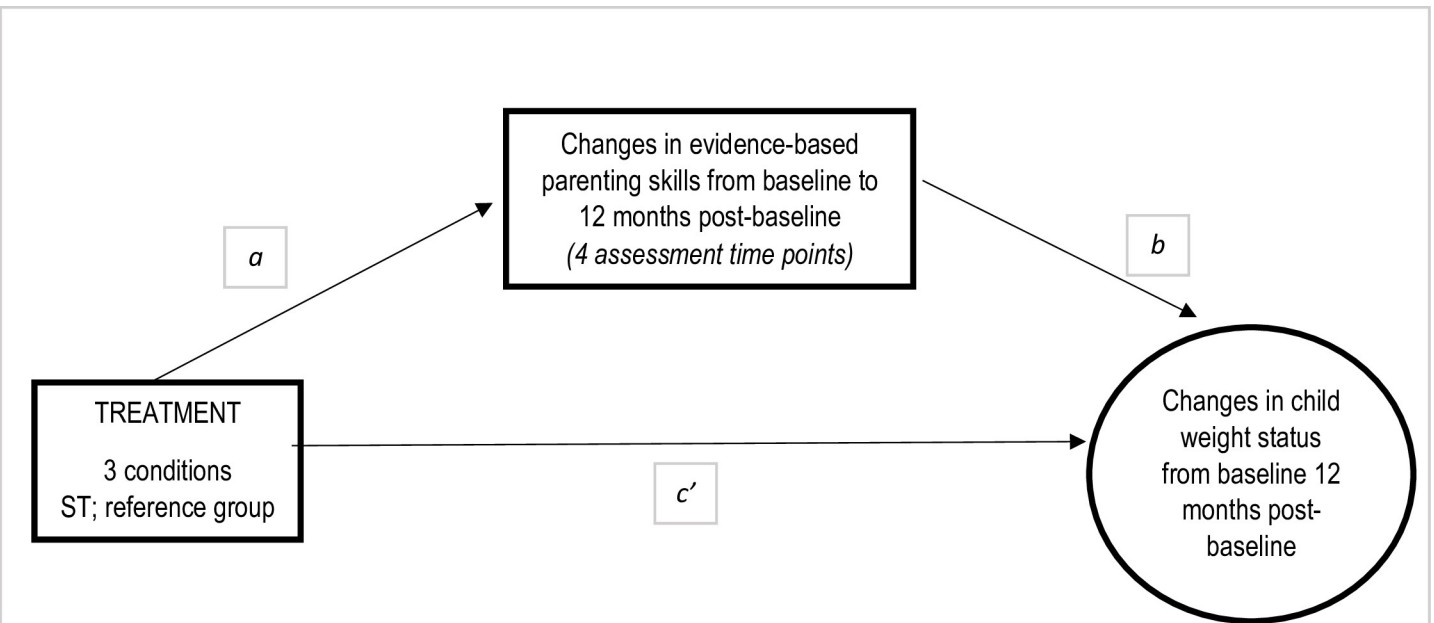

**Fig 2. Model specification for mediation, changes in parenting practices is the proposed mediator.** Treatment effects on the primary outcome (changes in child weight status at 12 months post-baseline) through changes in parenting practices (mothers' and fathers'). Pathway a: Direct treatment effect on changes in parenting practices (mediator). Pathway b: Direct effect of changes in parenting practices (mediator) on the primary outcome controlling for treatment. Pathway c': Direct treatment effect on the primary outcome adjusting for the proposed mediator. a*b: Indirect treatment effect on the primary outcome through the proposed mediator.

The reference group for mediation analysis was ST. The indirect effects (a*b) of PGB and PGNB on changes in the primary outcome through changes in parenting practices were computed and compared against the indirect effect of ST on the primary outcome. Significance was reached if the 95% CIs for these comparisons did not include '0' (Fig 2 illustrates the pathways a and b).

The software package IBM SPSS Statistics 24 was used for all statistical analyses and significance level was set to 0.05.

## Ethics approval

The study's ethics and consent procedures were approved by the Regional Ethical Board in Stockholm (approval numbers 2011/1329-31/4, 2012/1104-32, 2012/ 2005–32, 2013/486-32 and 2013/1628-31/2). In the clinical sample, caregivers provided written informed consent. In the school sample, data were collected anonymously, such that no informed consent was required.

## Results

### Study I

After the 12-item questionnaire was developed, its validity and reliability needed to be established. Table 1 shows the descriptive characteristics of the 558 parent-child dyads analyzed in the validation study. Children were on average 5 years old, and their mean BMI SDS was 0.5, which represents normal weight status. Mothers completed the majority of child questionnaires (81.7%). Most parents were born in Sweden (79.7%) and had a university degree (66.7%).

**Table 1. Descriptive characteristics of the total/merged sample (n = 558) in the validation study.**

|  | Total sample | School sample | Clinical sample |
|---|---|---|---|
|  | (n = 558) | (n = 427) | (n = 131) |
| **Child variables** | | | |
| Girl, n (%) | 292 (52.5) | 222 (52.2) | 70 (53.4) |
| Age in years, mean (SD) | 5.5 (1.0) | 5.5 (1.0) | 5.2 (0.7) |
| BMI SDS, mean (SD) | 0.5 (1.8) | -0.3 (1.2) | 2.9 (0.6) |
| Child obesity, n (%)[a] | 132 (26.6) | 1 (0.3) | 131 (100) |
| **Parent variables** | | | |
| Mothers, n (%) | 456 (81.7) | 349 (81.7) | 107 (81.7) |
| Age in years, mean (SD) | 38.6 (5.2) | 39.0 (6.5) | 37.4 (6.5) |
| BMI in kg/m$^2$, mean (SD) | 24.7 (4.6) | 23.6 (3.5) | 28.3 (5.7) |
| University education, n (%) | 368 (66.7) | 306 (72.3) | 62 (48.1) |
| Born in Sweden, n (%) | 441 (79.7) | 370 (87.3) | 71 (55.0) |

[a] 61 children in the school sample had missing weight status.

Table 2 presents the factor loadings of the PCA and Cronbach's alpha calculations, after three items were dropped from the 12-item questionnaire. The three items were 'If my child and I don't agree on something I wait with all discussion until the child has calmed down'

**Table 2. Cronbach's alpha and factor loadings of the questionnaire items on the two factors.**

| Item No. ‡ [a] | Item statements | Limit Setting (LS) [b] | Emotional Regulation (ER) [c] |
|---|---|---|---|
|  |  | *Cronbach's alpha* | |
|  |  | 0.80 | 0.70 |
|  |  | *Factor loadings* | |
| 5 | 'If my child and I disagree on something, we end up doing what my child wants' | 0.81 | - |
| 3 | 'I think it is difficult to say no to my child' | 0.78 | - |
| 7 | 'I can change my mind if my child throws a tantrum over something I have decided' | 0.75 | - |
| 1 | 'My child can make me change my mind to something I first said no to' | 0.73 | - |
| 12 | 'I think it is hard to set limits to my child' | 0.58 | - |
| 11 | 'If my child doesn't listen to me, I get frustrated' | - | 0.85 |
| 6 | 'If my child doesn't do what I say I find it hard to control my emotions' | - | 0.73 |
| 9 | 'If my child and I want different things, we end up falling out with each other' | - | 0.73 |
| 4 | 'How I handle my child's behavior depends on how I feel' | - | 0.55 |

Factor loadings to each factor are presented from highest to lowest. Factor loading threshold was set to 0.4. KMO Measure of sampling adequacy is acceptable (0.8) and Bartlett's Test of sphericity is significant.

‡ The order the items are shown is based on the factor they belong to (LS items appear first) and their factor loadings in descending order. Their numbering based on the order of appearance in the 12-item questionnaire is retained for identification purposes. Reversed coding was used for all items.

[a] Items 2, 8 and 10 (3 items) were dropped.

[b] Factor 1 (LS): variance explained 38.3%.

[c] Factor 2 (ER): variance explained 17.7%.

[b, c] Cumulative variance explained is 56%. Factors were chosen on the basis of correlations between their respective items (correlation matrices), scree plots (eigenvalues).

(item 2 in order of appearance in the questionnaire), 'My child listens to what I say' (item 8) and 'I think it's easy to get my child to think about something else if he/she starts nagging about something' (item 10). In two consecutive rounds of PCA, these items showed weak correlation with the rest of the items, and therefore did not have acceptable loadings (<0.4) onto any of the identified factors. After a third round of PCA, which included the remaining 9 items (9-item questionnaire), two factors were identified with acceptable reliability (Cronbach's alpha ≥0.7) (Table 2). Based on the content of their respective items (factor loadings >0.4), these factors were labelled 1) Limit Setting (LS), and 2) Emotional Regulation (ER). Therefore, the new questionnaire was titled "Emotions and Communication in Parenting (ECoP)". From now on, the abbreviations LS and ER will be used to describe the results, referring to parent limit setting and parent emotional regulation respectively.

On average, mean scores on LS were higher than scores on ER (4.03 *vs.* 3.51, p<0.05), yet both parenting practices were highly endorsed on the 5-point scale (Table 3). Ceiling effects, i.e. most parents picked high response categories, were shown for each item and the mean score in each practice (S1 Table; S1 Fig). Parents of children with obesity reported mostly high scores on LS items and small differences were reported regarding scores on ER. Differences in the individual items according to child weight status are provided in S2 Table. To examine construct validity, we examined correlations between mean scores for the two factors, as described above, and the CFQ and LBC (Table 3).

**Table 3. Correlations between limit setting/emotional regulation and the CFQ and LBC.**

| | Limit Setting (LS) | Emotional Regulation (ER) |
|---|---|---|
| **Child Feeding Questionnaire** [a] | Spearman's correlation coefficients | |
| Restriction | -0.15** | -0.07 |
| Pressure to eat | -0.03 | -0.1* |
| Monitoring | 0.11** | 0.09* |
| **Lifestyle Behavior Checklist** [b] | | |
| Overeating | -0.25** | -0.13** |
| Physical activity | -0.21** | -0.16** |
| Emotional correlates of being overweight | -0.14** | -0.03 |
| Misbehavior in relation to food | -0.29** | -0.19** |
| Screen time | -0.22** | -0.16** |
| Confidence Scale | 0.33** | 0.28** |
| | Mean (SD) | |
| Children without obesity | 4.1 (0.58) [‡] | 3.5 (0.64) |
| Children with obesity | 3.9 (0.76) [‡] | 3.6 (0.78) |
| Total sample (n = 558) | 4.03 (0.64) [¥] | 3.51 (0.70) [¥] |

Mean group differences in Limit Setting/Emotional Regulation between children with obesity and children without obesity (normal weight/overweight).

[‡] Significant difference between children with obesity and children without obesity by independent samples t-test (findings also confirmed using non-parametric Mann-Whitney U test), p<0.05.

[¥] Significant difference between mean scores for LS and ER in the total sample by paired samples t-test (findings also confirmed using non-parametric Wilcoxon signed rank sum test), p<0.001.

*p<0.05

**p<0.001.

[a] The items included are based on the findings from the validation study of the CFQ in Sweden [35].

[b] The items included are based on the findings from the validation study of the LBC in Sweden [33].

Both LS and ER correlated weakly with CFQ restriction, pressure to eat, and monitoring (no coefficient exceeded 0.15, absolute number). In particular, parents who reported higher scores on LS also reported higher scores in monitoring and lower scores in restriction. Parents reporting higher scores in ER reported lower scores in pressure to eat and higher scores in monitoring.

Regarding correlations with LBC factors, parents who reported high levels of LS and ER also reported lower scores in child problematic behaviors related to obesity (no coefficient exceeded 0.25, absolute number), and higher scores in confidence in tackling those problems (no coefficient exceeded 0.33, absolute number).

Parents of children with obesity reported lower scores in LS compared to parents of children without obesity (3.9 *vs.* 4.1, p<0.05). No group differences were found in ER.

## Study II

Table 4 shows the descriptive characteristics of the clinical sample. Child and parent variables did not differ across the three treatment groups, showing that the randomization was successful. Mean child age at baseline was 5 years, and child weight status assessed through BMI SDS was 2.97. The majority (60%) of mothers and fathers had foreign background, defined as being first- or second-generation migrants (with two parents born outside Sweden), while 40% of

**Table 4. Baseline characteristics and parental limit setting and emotional regulation in ML study.**

| | | Total sample | Parent program | | Standard treatment |
|---|---|---|---|---|---|
| | | N = 174 | *With* boosters n = 44 | *Without* boosters n = 43 | n = 87 |
| | N [a] | no. (%) or mean (SD) | no. (%) or mean (SD) | | no. (%) or mean (SD) |
| **Child variables** | | | | | |
| Girl | 174 | 98 (56.3) | 19 (43.2) | 23 (53.5) | 56 (64.4) |
| Living with both parents | 143 | 113 (79) | 25 (78.1) | 31 (81.6) | 57 (78.1) |
| First born | 147 | 72 (49) | 15 (41.7) | 21 (51.2) | 36 (51.4) |
| Age at baseline | 174 | 5.2 (0.78) | 5.2 (0.83) | 5.2 (0.86) | 5.3 (0.71) |
| BMI SDS at baseline | 174 | 3.0 (0.6) | 3.0 (0.5) | 3.0 (0.7) | 2.9 (0.6) |
| **Mother variables** | | | | | |
| Age | 139 | 36.6 (5.5) | 38 (5.1) | 36 (5.4) | 36 (5.7) |
| BMI | 141 | 28.1 (5.7) | 28.2 (6) | 29.1 (6.5) | 27.6 (5.1) |
| Foreign background | 145 | 89 (61.4) | 21 (63.6) | 21 (56.8) | 47 (62.7) |
| University degree | 143 | 58 (40.6) | 14 (42.4) | 15 (41.7) | 29 (39.2) |
| Limit Setting (LS) | 126 | 3.9 (0.8) | 3.9 (0.8) | 3.7 (0.8) | 3.9 (0.7) |
| Emotional Regulation (ER) [b] | 126 | 3.5 (0.8) | 3.1 (1.0) | 3.6 (0.8) | 3.7 (0.7) |
| **Father variables** | | | | | |
| Age | 124 | 39.8 (7.1) | 43 (7.9) | 39 (7.4) | 39 (6.3) |
| BMI | 126 | 29.4 (4.4) | 29.1 (4.2) | 30.02 (4.6) | 29.3 (4.5) |
| Foreign background | 130 | 75 (57.7) | 17 (54.8) | 21 (63.6) | 37 (56.1) |
| University degree | 128 | 49 (38.3) | 11 (36.7) | 12 (37.5) | 26 (39.4) |
| Limit Setting (LS) | 111 | 4.0 (0.7) | 4.1 (0.7) | 3.7 (0.9) | 4.1 (0.6) |
| Emotional Regulation (ER) | 112 | 3.8 (0.7) | 3.9 (0.6) | 3.7 (0.7) | 3.8 (0.8) |

[a] N size differs between variables due to missing data. Missing data on maternal and paternal LS and ER were more prevalent among mothers and fathers with a foreign background (parent and/or their parents born outside Sweden) and those without a university degree.

[b] Maternal ER at baselines differed across treatment groups, $F_{(2, 123)} = 4.51$, p = 0.01.

parents had a university degree. At baseline mothers and fathers reported high levels of LS/ER, 3.9/3.5 and 4.0/3.8 respectively (5-point scale). Nevertheless, mothers in the parent program with boosters reported lower levels of ER than the other groups.

Linear mixed models demonstrated that mothers across treatment groups did not differentially change their LS or ER practices (no significant group-by-time interaction). There was a significant effect of time (p = 0.001) whereby mothers in all groups increased their LS practices. By contrast, only fathers in ST increased their ER practices by 0.016 per month (p = 0.03), compared to fathers in PGNB and PGB (Fig 3 and S3 Table).

**Mediating effects of changes in evidence-based parenting practices on treatment effects.** No evidence was found for an indirect effect of treatment group on child weight status (primary outcome) through changes in parenting practices (S4 Table; 95% CI for the indirect pathway included 0).

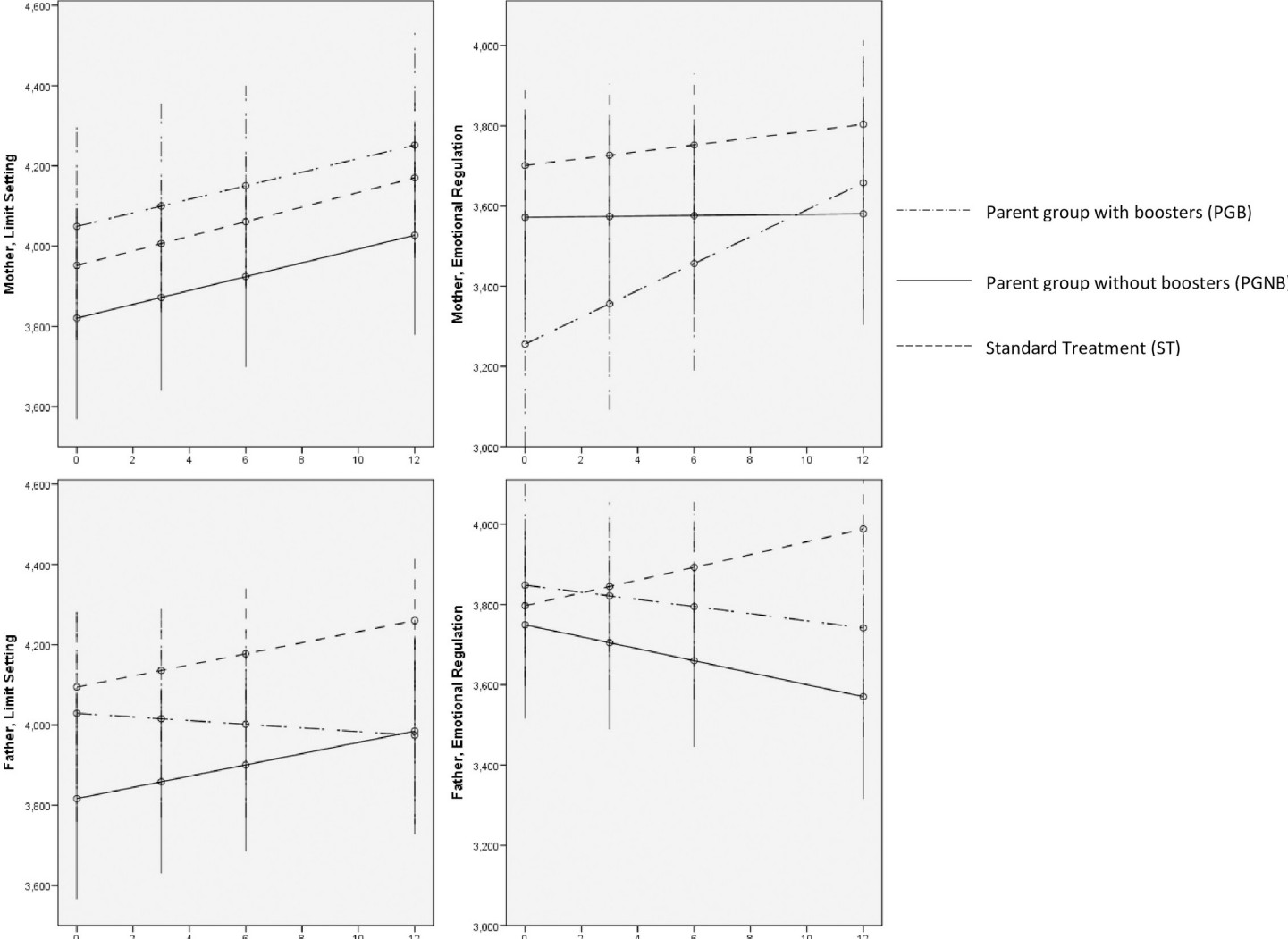

**Fig 3. Treatment effects over time in parenting practices among mothers and fathers.** Graphs are based on estimated marginal means of the linear mixed models fitted (holding time constant at 0, 3, 6, and 12 months). The horizontal axis represents Time (in months). A. Maternal Limit Setting: Group effect (p = 0.472), Time effect (p = 0.001), Group-by-Time (p = 0.993); B. Maternal Emotional Regulation: Group effect (p = 0.036), Time effect (p = 0.011), Group-by-Time (p = 0.075); C. Paternal Limit Setting: Group effect (p = 0.217), Time effect (p = 0.114), Group-by-Time (p = 0.237); D. Paternal Emotional Regulation: Group effect (p = 0.850), Time effect (p = 0.625), Group-by-Time (p = 0.026). All outcomes are assessed on the same Likert scale (from 1 to 5). The graphs illustrate different 1-point ranges within the Likert scale. Abbreviations: PGB: Parent group with booster sessions; PGNB: Parent group without booster sessions; ST: Standard Treatment.

## Discussion

This is the first study to investigate changes in parenting practices in an RCT of childhood obesity treatment involving families of preschoolers in Sweden. A new questionnaire on parenting practices, "Emotions and Communication in Parenting" (ECoP), was developed and validated to evaluate the evidence-based parenting practices included in the ML parenting program. The factor analysis of the new questionnaire yielded two dimensions of parenting practices. The first dimension assesses parents' limit setting and the second dimension assesses parents' ability to regulate their own emotions. Both dimensions of parenting showed expected correlations with feeding practices and with problematic child behaviors in relation to food, physical activity and obesity along with parental confidence in handling them. Using the ECoP to evaluate the RCT over a 12-month follow-up revealed that fathers in ST, on average, reported increased use of emotional regulation, while fathers in the parenting groups reported decreased use of this practice. By contrast, mothers across the study groups reported increased use of limit-setting practices to a similar degree. However, changes in parenting practices did not mediate the effect of obesity treatment on child weight status.

We expected the questionnaire to capture all the parenting practices addressed in the ML program (monitoring, positive involvement, limit setting, problem solving, emotional regulation, and encouragement). However, after consulting an expert group and parents, we found the final questionnaire identified two key dimensions of parenting, both of which allowed us to assess variations in parental practices: 1) parents' capacity to set limits to the child and 2) parents' capacity to regulate their own emotions in parenting situations. The findings may suggest that these two dimensions of parenting better differentiate between parents in the Swedish context (according to health professionals and parents), compared to the dimensions not represented in the final 9-item questionnaire. Interestingly, the two dimensions map onto the overarching dimensions that define parenting styles, i.e., responsiveness and demandingness [55–58]. Setting limits to one's child corresponds to providing clear and consistent boundaries and structure, while being able to regulate one's own emotions when interacting with the child allows parents to be responsive to the child's needs. Thus, it may be possible that the remaining dimensions of parenting which are practiced during the ML group sessions–i.e. positive involvement, monitoring, encouragement, and problem solving–act together to influence overarching dimensions of parenting (demandingness and responsiveness). This suggests that, at least in Sweden, all dimensions may be best assessed through parental limit setting and emotional regulation strategies. Overall, parents highly endorsed both limit setting and emotional regulation practices. Parents who scored highly on these positive aspects of parenting also scored highly on their confidence in handling problematic behaviors of the child, assessed through the LBC. Previous research has suggested the LBC Confidence Scale reflects 'a global measure of self-efficacy' [39], which aligns with the correlations found in our study. Accordingly, parents scoring highly in limit setting and emotional regulation reported fewer challenges concerning their children's problematic behaviors related to eating, physical activity and screen time.

As we have hypothesized, parenting practices correlated with specific feeding practices, providing evidence for the construct validity of the EcoP. In particular, parents who reported more effective limit setting strategies also reported higher monitoring and lower restriction of energy-dense foods. These contrasting patterns highlight possible differences between monitoring and restriction. While monitoring of child behaviors is a favorable form of control [32], overtly restricting access to energy-dense foods is coercive and not consistent with providing clear boundaries and structure in the home environment [59, 60]. However, lower restriction may also imply that parents limit access to these foods in the home environment, for example,

by not purchasing these foods, which decreases the need for restriction [61]. Thus, the correlations with limit setting may relate to an organized home environment with clear and consistent routines which support healthy behaviors [62, 63]. Moreover, parents that reported a higher capacity to regulate their emotions reported lower pressure to eat. This suggests that parents who find it easier to keep their emotions under control during challenging food situations with their children are also more likely to tune in to their children's will during feeding situations. Thus, parents may model self-regulation behaviors that enhance children's self-regulation of food and energy intake [64].

In study I, parents of children without obesity reported higher levels of limit setting strategies (though the total sample reported high scores in general), suggesting that favorable forms of control are associated with a healthy weight gain among young children [65], possibly through facilitating higher self-regulation in child eating [33]. Lack of differences in emotional regulation according to child weight status may reflect the overall high scores in the sample, or a universal responsiveness among parents of young children. Taken together, the findings confirm that the ECoP captures important aspects of parenting, alluding to structure in the home environment and consideration of child needs. Therefore, it is appropriate to use this questionnaire to evaluate changes in parenting practices in relation to feeding.

In study II, evaluation of changes in parenting practices in the ML study did not confirm our hypotheses. Mothers in the parenting program–with or without boosters–did not report increasing the parenting practices assessed by ECoP more than mothers in standard treatment during the 12-month follow-up. Fathers' treatment group allocation was associated with changes in their capacity to regulate their own emotions, but in an unexpected direction. Namely, fathers allocated to standard treatment increased their emotional regulation, in contrast to weaker effects among fathers in the parenting program, with and without boosters. This is surprising considering that children in standard treatment did not show significant weight loss [24]. The scarcity of evidence on the evaluation of parenting in obesity treatment, along with the heterogeneity of parenting assessments used in other studies, does not allow for direct comparisons with earlier RCTs. Nevertheless, our findings may align with an earlier prospective study showing that higher paternal responsiveness toward four-year-olds predicted higher risk of childhood overweight or obesity two years later, while maternal practices were not predictive of childhood obesity [12]. The authors highlighted that paternal responsiveness may reflect a more permissive parenting style whereby control and structure are reduced, thus conferring risk for greater weight gain over time. Our findings suggest that weaker effects on paternal emotional regulation in the parenting program (decreased use), compared to standard treatment (increased use), may have facilitated child weight loss, since limit-setting practices remained consistent over time. This is an important finding that highlights the unique contribution of fathers in obesity treatment, which has largely been overlooked in the literature [66]. It is possible a parent support program may be particularly beneficial for fathers, while mothers, who often have the main responsibility for feeding the child, may adjust their parenting practices through standard treatment without receiving the parenting component. However, within the family environment, both parents influence child health behaviors.

Whereas fathers' emotional regulation differed between standard treatment and the parenting program, mothers overall improved their ability to set limits, regardless of treatment condition. Similar findings were reported by Magarey et al. [67], who showed that integrating parenting components into childhood obesity treatment was as effective in increasing positive aspects of parenting as treatment focusing on lifestyle changes alone. These findings may imply that positive aspects of parental control increase during obesity treatment as part of applying lifestyle changes. However, this earlier study primarily involved mothers of older

children and used a different tool to assess parenting practices. On average, mothers in our study attended more sessions of the parenting program, compared to fathers. Based on our own clinical experience and reports from primary health care nurses, we can assume that the situation is similar in standard treatment, though we did not collect these data. Taken together, these findings suggest that parallel improvements in maternal parenting regardless of treatment condition and intensity (parenting program, with/without boosters and standard treatment), may be explained by higher maternal engagement in treatment overall.

An alternative explanation for our findings relates to the relative impact of mothers and fathers on child obesity-related behaviors. While both parents influence key obesity-related behaviors of the child, i.e. physical activity, screen time, and dietary intake, mothers seem to be more influential in the realm of child feeding [68, 69]. Lloyd, Lubans [68], in particular, demonstrated that specific practices of maternal limit setting related to higher energy intake from core foods (favorable), while paternal praise of child physical activity related to a lower count of daily steps (excessive praise may be perceived as coercion among children with overweight). In a recent publication, we showed that neither changes in maternal/paternal feeding practices nor changes in child food intake were plausible explanations of the clinically significant weight loss among children in the parenting program, especially in the group that received boosters, as compared to standard treatment [70]. All treatment conditions in the ML study highlighted physical activity along with healthy eating. Goal-setting strategies offered during the parenting program allowed each family to identify its own ways to implement changes in the home environment. Thus, it is possible that mothers in the parenting program, with and without boosters, assumed responsibility for implementing changes regarding child eating because they already had greater responsibility for feeding at baseline, while fathers intervened in addressing energy expenditure through increasing physical activity. An additional consideration is that booster sessions, which reinforced the program's messages throughout the 12-month follow up period, facilitated greater weight loss among children in this group. Assuming that fathers addressed children's physical activity more so than feeding, it is possible that fathers in the parenting program with boosters may have influenced child behaviors through physical activity–a variable we did not assess. More research is warranted on different practices endorsed by mothers and fathers and their effects on child obesity-related behaviors and weight status.

Despite treatment effects on parenting practices, there was no evidence that treatment effects on changes in child weight status (the primary outcome in the ML study) occurred through changes in parenting practices. This finding may relate to the alignment of the parenting outcomes (maternal and paternal) with the the content of ML parenting program. It has been suggested that more general aspects of parenting, such as limit setting and emotional regulation, are not directly related to child outcomes [71, 72], and, therefore, weak or no associations can be expected. However, parenting practices may influence associations between specific feeding practices and child outcomes. For example, Sleddens, Kremers [73] showed that when parents encouraged children to taste and enjoy their meals, within a structured and positive parenting context, children decreased their unhealthy dietary intake. These favorable associations with child dietary intake were not present in less positive parenting contexts. Taken together, these findings suggest that parenting practices may affect child weight outcomes through complex pathways involving diverse parent and child variables. It is, therefore, important to examine such pathways in obesity treatment and evaluate how changing parenting practices may influence the effects of feeding and physical activity practices on child outcomes.

## Strengths and limitations

The present study has several strengths and limitations. A notable strength is the development and validation of a new questionnaire to assess parenting practices in childhood obesity treatment, ECoP, which followed a comprehensive structured approach as recommended [34, 35, 74]. The greater participation of mothers in the validation study (almost 82%) compared to fathers may relate to mothers' primary feeding responsibilities [69]. However, recent studies have called for the inclusion of fathers in parenting research, and future research should evaluate similarities and differences between mothers' and fathers' parenting and feeding practices [56, 68, 75, 76]. Although the sample in the validation study was well-educated and with a lower proportion of parents with a foreign background compared to the general population, the response rate (46%) was similar to earlier studies focusing on parenting instruments [43, 77]. The addition of the clinical sample yielded a more heterogeneous population, increasing the external validity of the study. Moreover, the development of the parenting questionnaire was informed by the parenting components addressed in the ML parenting program. This allowed for the alignment of treatment content with the evaluation of treatment effectiveness, and is an important step in understanding key mechanisms in obesity treatment among preschoolers. However, further research is needed to establish the relevance of the ECoP in relation to child developmental outcomes other than childhood obesity, which has been the focus in the present paper. In addition, more than half of the parents participating in the ML study were first- or second-generation migrants. The sample had an overall lower educational attainment than the more homogeneous and well-educated samples dominating research in this field. This addresses widely recognized limitations in childhood obesity studies, which tend to focus on sociodemographically homogenous samples [78], and thereby allows us to include a wider variety of manifestations of parenting practices [79]. However, cross-cultural comparisons of parenting practices and their possible infuence on child outcomes were beyond the scope of this study. Future research should highlight and thoroughly examine the moderating role of migrant background in the effectiveness of childhood obesity treatment, as shown in relation to child weight status in the ML study [24], but also in relation to secondary outcomes, such as parenting practices. The study's primary limitation is that power calculations were based on the primary outcome of the ML study (changes in child weight status over a 12-month follow-up). Therefore, we may not have had enough power to detect meaningful differences in parenting practices between treatment groups. However, we were still able to demonstrate the importance of including both mothers and fathers in childhood obesity studies, to understand their potentially different effects on treatment outcomes.

## Conclusions

This is the first study to investigate how changes in evidence-based parenting practices may influence the outcomes of obesity treatment for preschool-age children. The study included the development and validation of a new questionnaire "Emotions and Communication in Parenting" (ECoP). This questionnaire allowed us to assess the parenting practices addressed in an RCT, the ML study, that compared a parenting program with and without boosters, and standard treatment. Based on the validation of this questionnaire, two relevant dimensions of parenting emerged: 1) parents' ability to set limits to the child, and 2) parents' capacity to regulate their own emotions. Both dimensions of parenting correlated with specific practices on validated questionnaires: parental feeding practices (on the CFQ) and challenging child behaviors related to eating and obesity (on the LBC). The evaluation of the treatment program showed that, during the 12-month follow-up, the measured parenting practices did not mediate changes in child weight status. However, fathers' and mothers' practices changed in

different ways over the course of obesity treatment. Fathers in the parenting program (with and without boosters) decreased their use of emotional regulation, compared to fathers in standard reatment who increased theirs. By contrast, mothers in both the parenting program and standard treatment increased their use of limit setting strategies to a similar degree. Our findings highlight that fathers' practices need to be considered when developing and evaluating childhood obesity treatment. Future research should investigate more closely the specific practices mothers and fathers employ, and evaluate how these might influence child behaviors and child weight status.

## Supporting information

**S1 Table. Distribution of response categories of all items in the parenting questionnaire in the validation study (study I).**
(DOCX)

**S2 Table. Mean differences regarding questionnaire items between children with normal weight/overweight and children with obesity (study I).**
(DOCX)

**S3 Table. Effects of treatment group, time and group-by-time interaction on the four parenting practices outcomes-maternal limit setting and emotional regulation, and paternal limit setting and emotional regulation.**
(DOCX)

**S4 Table. Mediation model.** Path coefficients to examine the mediating effect of parenting practices (mothers' and fathers') on treatment effects on the primary outcome (changes in child weight status at 12 months post-baseline).
(DOCX)

**S1 Fig. Distribution of mean scores in limit setting and emotional regulation in the validation study (study I).** Mean scores in both parenting practices were higher than average.
(DOCX)

## Acknowledgments

We want to thank all participating families, child health care and school nurses, and all personnel involved in the standard treatment offered in the pediatric outpatient clinics. We also thank Jonna Nyman, Mahnoush Etminan Malek, Karin Nordin, Kathryn Lewis Chamberlain, Jan Ejderhamn, Philip A. Fisher, Patricia Chamberlain, and Claude Marcus who were involved in design or in data collection in the More and Less Study.

## Author Contributions

**Conceptualization:** Maria Somaraki, Anna Ek, Sofia Ljung, Paulina Nowicka.

**Formal analysis:** Maria Somaraki, Anna Ek, Karin Eli, Veronica Mildton.

**Funding acquisition:** Paulina Nowicka.

**Methodology:** Maria Somaraki, Anna Ek, Karin Eli, Sofia Ljung, Veronica Mildton, Pernilla Sandvik, Paulina Nowicka.

**Project administration:** Anna Ek, Sofia Ljung, Paulina Nowicka.

**Resources:** Veronica Mildton, Paulina Nowicka.

**Supervision:** Pernilla Sandvik, Paulina Nowicka.

**Writing – original draft:** Maria Somaraki, Karin Eli.

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
