## [Decision Letter · Decision Letter 0]

8 Feb 2021

PONE-D-20-28940

Parenting and childhood obesity: Validation of a new questionnaire and evaluation of treatment effects during the preschool years

PLOS ONE

Dear Dr. Nowicka,

Thank you for submitting your manuscript to PLOS ONE. After careful consideration, we feel that it has merit but does not fully meet PLOS ONE’s publication criteria as it currently stands. Therefore, we invite you to submit a revised version of the manuscript that addresses the points raised during the review process.

We look forward to receiving your revised manuscript.

Kind regards,

John William Apolzan, PhD

Academic Editor

PLOS ONE

Journal Requirements:

2. Please provide the registration information for the clinical trial associated with this study.

Additional Editor Comments:

Please respond to all reviewer comments but particularly ensure the statistical questions are adequately addressed.

Reviewers' comments:

Reviewer's Responses to Questions

**Comments to the Author**

1. Is the manuscript technically sound, and do the data support the conclusions?

Reviewer #1: Partly

Reviewer #2: Partly

Reviewer #3: Yes

2. Has the statistical analysis been performed appropriately and rigorously? 

Reviewer #1: I Don't Know

Reviewer #2: Yes

Reviewer #3: Yes

3. Have the authors made all data underlying the findings in their manuscript fully available?

Reviewer #1: Yes

Reviewer #2: No

Reviewer #3: No

4. Is the manuscript presented in an intelligible fashion and written in standard English?

Reviewer #1: No

Reviewer #2: Yes

Reviewer #3: Yes

5. Review Comments to the Author

Reviewer #1: Important note: This review pertains only to ‘statistical aspects’ of the study and so ‘clinical aspects’ [like medical importance, relevance of the study, ‘clinical significance and implication(s)’ of the whole study, etc.] are to be evaluated [should be assessed] separately/independently. Further please note that any ‘statistical review’ is generally done under the assumption that (such) study specific methodological [as well as execution] issues are perfectly taken care of by the investigator(s). This review is not an exception to that and so does not cover clinical aspects {however, seldom comments are made only if those issues are intimately / scientifically related & intermingle with ‘statistical aspects’ of the study}. Agreed that ‘statistical methods’ are used as just tools here, however, they are vital part of methodology [and so should be given due importance].

COMMENTS: Your ABSTRACT is well drafted but assay type. Please note that it is preferable [refer to item 1b of CONSORT checklist 2010: Structured summary of trial design, methods, results, and conclusions] to divide the ABSTRACT with small sections like ‘Objective(s)’, ‘Methods’, ‘Results’, ‘Conclusions’, etc. which is an accepted practice of most of the good/standard journals [including PLOS-ONE]. It will definitely be more informative then, I guess, whatever the article type may be.

Unfortunately, there are several questions [naturally arising in mind] about this article. Few important/vital ones [not possible to cover all] are described/given below. First let us refer to lines 198-202, section on ‘Sample size’ [as vital for present statistical review] which quotes reference by Kleber et al., 2009 {‘Power calculations were based on the primary outcome, child BMI SDS (Body Mass Index Standard Deviation Score) (Kleber et al., 2009)}. Are you referring to reference number 19? But it is said in line 30 that their study (reference 19) is ‘without a control condition’ then how can ‘power calculations are based’ on this study or do you want to say that ‘the primary outcome, child BMI SDS (Body Mass Index Standard Deviation Score)’ this term is used for first time by them or the term is taken from this reference? Clarify the purpose of quoting the article. Note that “Kleber” is second author of paper quoted as 19 and instead of [19], quoting (Kleber et al., 2009) is not correct at all. Please follow the standard practice of quoting reference(s).

Further, it is said that ‘…. adjusting for a dropout rate of 21%’. Are following references (namely Ek, Chamberlain, et al., 2015; West et al., 2010) are for the (odd) figure of 21%? Account given in section on ‘Sample size’ (lines 198-202) ‘Seventy-five children needed to be included in each of the treatment approaches (ML program and standard treatment) in order to identify a difference in BMI SDS between the groups at 12-months post-baseline’ is/are inadequate [not sufficient at all] and therefore, of no use { to identify what amount of difference in BMI SDS?}.

Though measures/tools used are appropriate, most of them [example: Child Feeding Questionnaire (CFQ), the Lifestyle Behaviour Checklist (LBC), etc.] yield data that are in [at the most] ‘ordinal’ level of measurement [and not in ratio level of measurement for sure {as the score two times higher does not indicate presence of that parameter/phenomenon as double (for example, a Visual Analogue Scales VAS score or say ‘depression’ score)}]. Then application of suitable non-parametric test(s) is/are indicated/advisable [even if distribution may be ‘Gaussian’ (i.e. normal)].

Now refer to lines 228-234

{Statistical Analyses - Sub study I: Differences between subsamples were examined using independent samples t-test for continuous variables and chi-squared test for categorical variables. Background characteristics were compared across the school and clinical samples using one-way Analysis of Variance (ANOVA) (for continuous variables) and chi-square test (for categorical variables).}

Use Mann-Whitney U test in place of independent samples t-test. Note that we use Analysis of Variance (ANOVA) [for continuous variables] to compare three or more groups {not to compare two groups]. You seem to compare two groups [the school and clinical samples] only. Following note is from one standard text-book on Biostatistics:

When only two groups are to be compared {ex. Women and Men}, we use ‘t’ test for two independent groups [non-parametric equivalent to unpaired ‘t’ test is Mann-Whitney ‘U’ test] and not ANOVA ‘F’ Although ‘F’ and ‘t’ are mathematically related/equivalent [square of ‘t’ is exactly equal to ‘F’ if (mistakenly) calculated for two groups], logic/philosophy (and so underlying assumptions) behind their development and algorithms used for estimation of test statistic are different. Mind you that they are applicable in different situations.

Choice of test will not depend on ‘Type of variable’ or ‘Level of measurement’ of parameter/background characteristic [line 231 or 256] under consideration. {Non-parametric equivalent of one-way ANOVA ‘F’ is ‘Friedman’s test’}. If treatment group are now [i.e. your are considering] three conditions [as PGB: Parent group with booster sessions; PGNB: Parent group without booster sessions; ST: Standard Treatment], clarify. Kindly remember that this is a scientific/academic document and so all details should be clearly communicated.

In ‘Abstract’ you say “A merged school/clinical sample (n=558, 82% mothers) was used for the factorial and construct validation of a new parenting questionnaire. Changes in parenting were evaluated using data from the More and Less Study, a randomized controlled trial (RCT) with 174 children (mean age=5 years, mean BMI-Z =3.0) comparing a parent support program (with and without booster sessions) and standard treatment.”. Further, you say “We administered the new questionnaire to the RCT participants.”. Is not that confusing? Where and when this RTC (line 39-40: an RCT for obesity treatment among preschoolers, the More and Less study (ML study) [23].) was conducted? As said in lines 54-6 [The evaluation includes two stages: performing a validation study on a new questionnaire on parenting practices (Sub study I) and assessing parenting practices using data from the ML study RCT (Sub study II).] it seems that you have used entirely different samples for these two stages {Sub study I & Sub study II}. Is that true? It is not clear ‘whether n=558 or n=174’ for this study [how can different stages of the same can use different samples?]. Again, remember that this is a scientific/academic document and so all details should be clearly communicated. Do not confuse readers. I found many things confusing in this article. Please do not take readers for granted {that they will understand what is there in your mind}. It may be 100% correct, but need to (should) be communicated clearly.

Refer to Table 3 [Correlations between Limit Setting/Emotional Regulation and the CFQ and LBC. Mean group differences in Limit Setting/Emotional Regulation between children with obesity and children without obesity (normal weight/overweight)]. In this context note that [mainly because you have used ‘n.s.’ in this table],

Statistical test usually used to assess significance of Pearson’s ‘Correlation coefficient (r)’ is ‘t’ [where t = { r � [(n-2) / (1-r2)] }for df=n-2, n is sample size] and here Ho is that the population/standard value of ‘r’ is zero. You need r=0.878 to be significant at 5% when n=5 but you need r=0.273 if n=50 & you need only r=0.088 if n=500. ‘P-value’ heavily depends on sample size. Therefore, it is customary to use the (available in most text books on ‘Biostatistics’ or on ‘www/net’) following guidelines for interpreting positive or negative correlations (and do not rely only on corresponding ‘P’-value but also consider an absolute value of ‘Correlation coefficient’). [This argument is equally applicable to non-parametric Spearman’s ‘Correlation coefficient (ρ)’ as well.]

There are two more questions regarding Table three.

1. You have tested mean group differences in ‘Limit Setting & Emotional Regulation’ between children with obesity and children without obesity separately by ‘Individual samples t-test’ which is perfectly alright but why ‘Paired samples t-test’ is applied for TOTAL? Is that correct (are you dealing with pairs now)? How? Please explain {this question is due to footnote}.

2. Why references are quoted in footnote [a Validation study of the CFQ in Sweden [35], b Validation study of the LBC in Sweden [33]]. This implies that ‘r’ values are reported from these references. Is that so? Please explain.

I refrain from giving adverse comments on many other points in manuscript however, I definitely feel [am almost sure] that this study has potential. Presentation of (hard achieved) material is poorly done. Re-drafting avoiding confusion is recommended.

Reviewer #2: 2. The authors used appropriate statistical methods to address their research questions. However, the rigor of these analyses is unclear (e.g., were models adjusted for relevant covariates). I have addressed this in my uploaded attachment for authors.

Reviewer #3: Manuscript Number: PONE-D-20-28940

Title: Parenting and childhood obesity: Validation of a new questionnaire and evaluation of treatment effects during the preschool years

Comments to the Authors

This manuscript concerns the validation of a new questionnaire about the evaluation of mothers’ and fathers’ parenting practices in relation to child weight status during a 12-month childhood obesity treatment trial. The manuscript is well-written, however, there are some points that need clarifications or changes. Please see my comments below.

Abstract

In the abstract, the authors stated that the validation of the new questionnaire (12 items; responses on a 5-point Likert scale) revealed two dimensions of parenting (Cronbach’s alpha ≥0.7). However, in the section of the results, they stated that the PCA and Cronbach’s alpha calculations were done after three items were dropped from the 12-item questionnaire. I wonder if the final questionnaire contains 12 or 9 questions. This is a point that should be clarified and corrected.

Introduction

My major objection is the necessity for the construction of a new tool. At the end of the Introduction the authors just mentioned that “To evaluate parenting practices, a valid tool was needed [27]”. Then they continued with “Therefore, a new parenting questionnaire, which could match the components of the ML parenting program, was developed”. From this last sentence, it is deduced that this questionnaire was constructed just to meet the needs of the ML program. This point needs to be clarified as if this is the case, then the generalizability of the new tool is questioned.

Materials and methods

The literature search procedure needs more description. The authors should report all databases they searched. Furthermore, it is not clear whether only 14 questionnaires were identified or were more tools and the authors selected these 14?

Some comments concerning the school sample:

By what sampling method were the fifteen schools and thirty preschools selected?

In the school sample, almost half of the parents (431 out of 931) returned completed questionnaires. The authors should comment on whether this has any effect on the results. Do the characteristics of the parents who did not respond differ from those of the parents who participated?

Results

The “Child obesity” variable displayed in Table 1 has not been defined. How were the children categorized as obese and not obese? In general, all variables used in this study should be mentioned and explained.

In line 336 the authors stated that on average, mean scores on LS were higher than scores on ER (4.03 vs. 3.51). They should also provide the corresponding p-value.

Discussion

In lines 409 – 413, the authors stated that although they expected the new questionnaire to capture all the parenting practices addressed in the ML study, it captured only two of the dimensions of parenting (parents’ capacity to set limits to the child and parents’ capacity regulate their 413 own emotions in parenting situations). I would expect to see some possible explanations for this.

6. PLOS authors have the option to publish the peer review history of their article (what does this mean?). If published, this will include your full peer review and any attached files.

Reviewer #1: No

Reviewer #2: No

Reviewer #3: No

---

## [Author Response · Author response to Decision Letter 0]

18 Jun 2021

We thank the editor for giving us the opportunity to improve our manuscript. We have responded to all the reviewers’ comments, which have been constructive and very helpful. The response letter is provided along with all the revised files.

---

## [Decision Letter · Decision Letter 1]

26 Aug 2021

Parenting and childhood obesity: Validation of a new questionnaire and evaluation of treatment effects during the preschool years

PONE-D-20-28940R1

Dear Dr. Nowicka,

We’re pleased to inform you that your manuscript has been judged scientifically suitable for publication and will be formally accepted for publication once it meets all outstanding technical requirements.

Kind regards,

John W. Apolzan, PhD

Academic Editor

PLOS ONE

Reviewers' comments:

Reviewer's Responses to Questions

**Comments to the Author**

1. If the authors have adequately addressed your comments raised in a previous round of review and you feel that this manuscript is now acceptable for publication, you may indicate that here to bypass the “Comments to the Author” section, enter your conflict of interest statement in the “Confidential to Editor” section, and submit your "Accept" recommendation.

Reviewer #1: All comments have been addressed

Reviewer #2: All comments have been addressed

2. Is the manuscript technically sound, and do the data support the conclusions?

Reviewer #1: (No Response)

Reviewer #2: Yes

3. Has the statistical analysis been performed appropriately and rigorously? 

Reviewer #1: (No Response)

Reviewer #2: Yes

4. Have the authors made all data underlying the findings in their manuscript fully available?

Reviewer #1: (No Response)

Reviewer #2: (No Response)

5. Is the manuscript presented in an intelligible fashion and written in standard English?

Reviewer #1: (No Response)

Reviewer #2: Yes

6. Review Comments to the Author

Reviewer #1: COMMENTS: Most of the comments made on earlier draft(s) by me (and hopefully by other respected reviewers also) were/are attended very positively. It is very good that [as you said in response to my fifth comment] the findings and conclusions drawn based on the nonparametric tests are largely the same as before, however, note that one should apply correct/indicated methods/tests always.

Though I am not fully satisfied (because the study has more potential than appears/apparent from the presentation), I recommend the acceptance as the manuscript now has achieved acceptable level of our journal, in my opinion.

Reviewer #2: My concerns have been addressed. I have no further comments. Great job to the Authors for their hard work!

7. PLOS authors have the option to publish the peer review history of their article (what does this mean?). If published, this will include your full peer review and any attached files.

Reviewer #1: **Yes: **Dr. Sanjeev Sarmukaddam

Reviewer #2: No

---

## [Editor Report · Acceptance letter]

15 Sep 2021

PONE-D-20-28940R1 

Parenting and childhood obesity: Validation of a new questionnaire and evaluation of treatment effects during the preschool years 

Dear Dr. Nowicka:

I'm pleased to inform you that your manuscript has been deemed suitable for publication in PLOS ONE. Congratulations! Your manuscript is now with our production department. 

Kind regards, 

on behalf of

Dr. John W. Apolzan 

Academic Editor

PLOS ONE